# Topological–chiral magnetic interactions driven by emergent orbital magnetism

S. Grytsiuk [1]*, J.-P. Hanke [1], M. Hoffmann [1], J. Bouaziz[1], O. Gomonay[2], G. Bihlmayer [1], S. Lounis [1], Y. Mokrousov [1,2] & S. Blügel [1]

Two hundred years ago, Ampère discovered that electric loops in which currents of electrons are generated by a penetrating magnetic field can mutually interact. Here we show that Ampère's observation can be transferred to the quantum realm of interactions between triangular plaquettes of spins on a lattice, where the electrical currents at the atomic scale are associated with the orbital motion of electrons in response to the non-coplanarity of neighbouring spins playing the role of a magnetic field. The resulting topological orbital moment underlies the relation of the orbital dynamics with the topology of the spin structure. We demonstrate that the interactions of the topological orbital moments with each other and with the spins form a new class of magnetic interactions — topological–chiral interactions — which can dominate over the Dzyaloshinskii–Moriya interaction, thus opening a path for realizing new classes of chiral magnetic materials with three-dimensional magnetization textures such as hopfions.

[1] Peter Grünberg Institut and Institute for Advanced Simulation, Forschungszentrum Jülich and JARA, 52425 Jülich, Germany. [2] Institute of Physics, Johannes Gutenberg University Mainz, 55099 Mainz, Germany. *email: s.grytsiuk@fz-juelich.de

Exotic magnetic textures with particle-like properties[1–6] offer great potential for innovative spintronic applications[7] and brain-inspired computing[8,9]. Magnetic skyrmions, two-dimensional (2D) localized solitons, are a prominent realization of chiral spin structures, first observed in the material class of non-centrosymmetric B20 bulk compounds[1]. The potential of spintronic applications would change fundamentally if the line of thought could be continued to the emergence of three-dimensional (3D) localized magnetic solitons, e.g., hopfions[10–12]. Recently, a 3D lattice of 3D magnetic textures on the nanometer scale was observed in the B20-type cubic chiral magnets MnGe[13,14]. Despite the strong interest in this magnet, a complete theoretical model for the underlying magnetic interactions is remarkably elusive until now. While, for instance, the basic magnetic properties of the 2D skyrmions are determined by an intricate competition involving the Heisenberg exchange and the chiral relativistic Dzyaloshinskii–Moriya interaction[15,16] (DMI), such models fail to explain the 3D-magnetic texture observed in MnGe[17].

The 3D magnetization textures of 2D skyrmions gives rise to a scalar spin chirality, a driving force behind a plethora of macroscopic phenomena. Examples are the topological Hall effect[18,19] or a finite topological orbital moment (TOM)[20–25], which can both serve as experimental fingerprints of skyrmions. Texture-induced contributions to these macroscopic phenomena were also predicted in frustrated magnets[26,27], where they originate from the non-trivial spin topology associated with the real-space configuration of magnetic moments $\mathbf{S}_i$ as reflected by the scalar spin chirality $\chi_{ijk} = \mathbf{S}_i \cdot (\mathbf{S}_j \times \mathbf{S}_k)$. Although the net spin magnetization might vanish, the symmetry of these chiral systems allows for lowering the energy by preferring orbital currents of specific rotational sense[26,28]. As a consequence, the motion of the electron in the complex magnetic background manifests itself in the finite TOM without any reference to typical relativistic mechanisms.

Instead, this response is usually ascribed to an emergent magnetic field $\mathbf{B}^{\text{eff}}$ that roots in the non-coplanar spin texture, giving rise to spontaneous orbital currents, see Fig. 1. While these non-relativistic currents have been so far largely overlooked, only lately, the perception that they could contribute to the energetics of spin systems is nascent.

Here, based on microscopic arguments and a systematic total-energy expansion, we discover a conceptually new class of chiral interactions between spins on triangular plaquettes, which originates from the TOM of electrons. We refer to these interactions as topological–chiral interactions, favouring the emergence of non-coplanar magnetic structures with scalar spin chirality of specific sign even without an external magnetic field, either in the ground state or as a result of thermal fluctuations[29]. The first type of topological–chiral interactions is the rotation-invariant chiral–chiral interaction (CCI), which in its general form corresponds to the interaction between pairs of topological orbital currents in a magnet, just in analogy to Ampère's observation that the force between wires can be described by the effective interaction of currents. The second type of topological–chiral interactions is the rotationally anisotropic spin–chiral interaction (SCI), which arises as a result of a direct coupling between the TOM and local spins, mediated by the spin-orbit interaction. We uncover the importance of the discovered topological-chiral interactions for the energetics of spin systems by explicit first-principles calculations in B20 magnets. Finally, if the emerging magnetic textures can be represented by continuous magnetization fields, we show that systems described by the conventional Heisenberg and the chiral–chiral interactions form a physical realization of the Faddeev model[30] with hopfion solutions[10–12]. This signifies the key role of the topological-chiral interactions in triggering the formation of 3D magnetic solitons without the assistance of an external magnetic field.

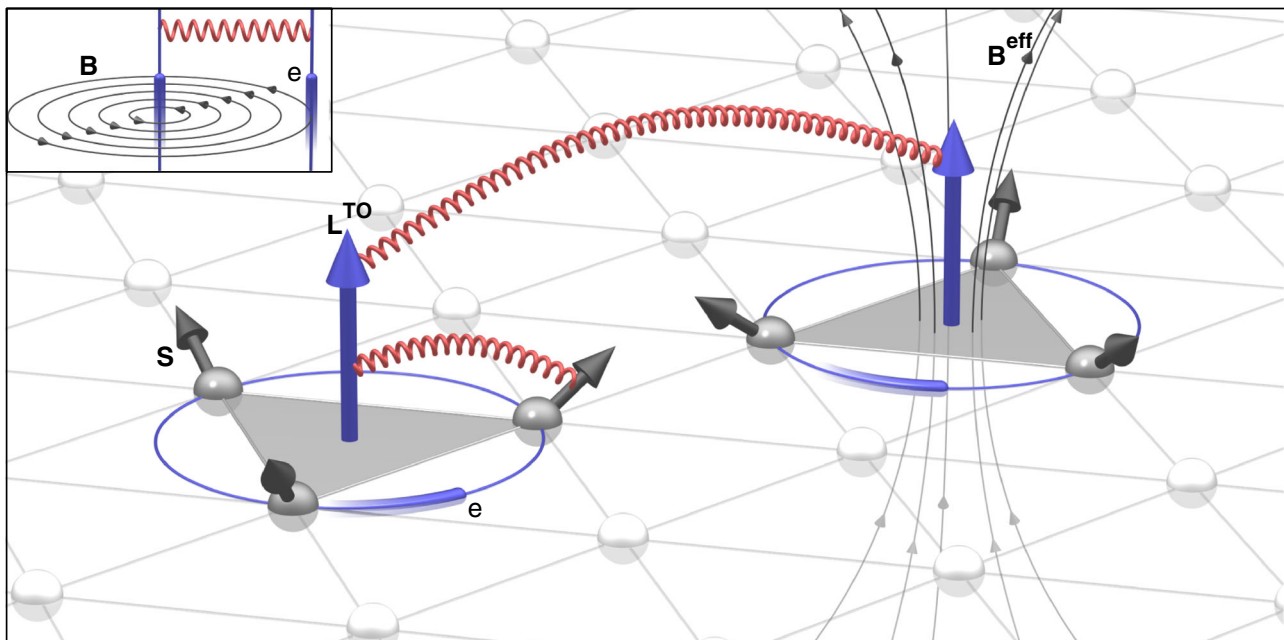

**Fig. 1 Topological–chiral interactions.** In a magnet exhibiting a non-coplanar spin arrangement (black arrows), the local scalar spin chirality between the triplets of spins can be interpreted as an effective magnetic field $\mathbf{B}^{\text{eff}}$ which gives rise to a so-called topological orbital moment $\mathbf{L}^{\text{TO}}$ (TOM, blue arrows) generated by the orbital current of electrons hopping around the triangle. In analogy to Ampère's law shown as inset, the topological orbital currents, generated by different plaquettes of spins, can interact with each other, giving rise to the first type of topological-chiral interactions — the rotation-invariant chiral–chiral interaction (CCI). The second type of topological-chiral interactions — the spin–chiral interaction (SCI) — corresponds to the coupling between TOM and the local spins, mediated by the spin-orbit interaction.

## Results

**Topological orbital moment.** We briefly elucidate the concept of emergent orbital currents due to the motion of electrons through a non-collinear magnetic structure[26,28]. In the absence of the spin-orbit interaction (SOI), hopping in a non-coplanar spin texture with finite chirality $\chi_{ijk} = \mathbf{S}_i \cdot \left(\mathbf{S}_j \times \mathbf{S}_k\right)$, where $\mathbf{S}$ are direction vectors of classical spin moments located at sites $i, j, k$, is equivalent to the electron dynamics in a coplanar spin background but in the presence of a fictitious magnetic field $B^{\mathrm{eff}} \propto \chi_{ijk}$ directed along $\boldsymbol{\tau}_{ijk}$[31], which gives rise to a Berry phase[26]. Here, the vector $\boldsymbol{\tau}_{ijk} \propto (\mathbf{R}_j - \mathbf{R}_i) \times (\mathbf{R}_k - \mathbf{R}_i)$ is the surface normal of the oriented triangle spanned by the lattice sites $\mathbf{R}_i$, $\mathbf{R}_j$, and $\mathbf{R}_k$ at which spin moments are placed. Acting as an effective magnetic field, the complex spin topology of chiral systems thereby allows for ground-state currents of specific rotational sense that manifest in spontaneous orbital properties of the electrons. This microscopic mechanism induces a topological orbital moment (TOM) that stems from the scalar spin chirality without any reference to relativistic origins, as predicted for several situations ranging from non-collinear 3Q-states to large-scale skyrmions[20–25]. In the limit of vanishing effective magnetic field, i.e., in a situation of small spin chirality, the spontaneous TOM is directly proportional to $B^{\mathrm{eff}}$ with the proportionality factor given by the topological orbital susceptibility of the system[25].

As illustrated in Fig. 1, the TOM which roots in emergent orbital currents around a given spin triangle can be linked phenomenologically to the triangle normal and the corresponding scalar spin chirality via $\mathbf{L}_{ijk}^{\mathrm{TO}} = \kappa_{ijk}^{\mathrm{TO}} \chi_{ijk} \boldsymbol{\tau}_{ijk}$, where $\kappa_{ijk}^{\mathrm{TO}}$ is the local topological orbital susceptibility. Using these individual orbital moments, we quantify the local TOM at the $i$th atomic site by taking into account that each spin participates in several triangles:

$$\mathbf{L}_i^{\mathrm{TO}} = \sum_{(jk)} \mathbf{L}_{ijk}^{\mathrm{TO}} = \sum_{(jk)} \kappa_{ijk}^{\mathrm{TO}} \chi_{ijk} \boldsymbol{\tau}_{ijk} , \qquad (1)$$

where the sum over $(jk)$ is restricted to triangles including the $i$th magnetic atom in the unit cell. In the following, we demonstrate that these orbital properties are key entities of a novel class of chiral exchange interactions, which we refer to as topological–chiral interactions imprinting on the ground-state energetics of spin systems. They appear in two distinct realizations.

**Chiral–chiral interaction.** When following the rationale of Ampère's law down to the quantum scale, it is intuitively clear that the mutual interaction between emergent orbital currents due to the scalar spin chirality contributes to the total energy of chiral systems. While we derive this non-relativistic interaction explicitly in Supplementary Note 1 from a systematic total-energy expansion based on multiple scattering theory (see Methods), here, we motivate its final form in terms of macroscopic physical properties. As depicted in Fig. 1, in the absence of SOI, the resulting chiral–chiral interaction (CCI) correlates emergent orbital currents occurring on spin triangles, just in analogy to Ampère's force law correlating electrical currents. The most dominant contribution to the total energy is expected to originate from currents around the same plaquette such that the local part of the CCI can be interpreted as the orbital Zeeman interaction $\mathbf{L}^{\mathrm{TO}} \cdot \mathbf{B}^{\mathrm{eff}}$ of the spontaneous TOM with the emergent magnetic field of the non-coplanar spin background:

$$
\begin{aligned}
E^{\mathrm{CC}} &= -\frac{1}{2} \sum_{i(jk)} \kappa_{ijk}^{\mathrm{CC}} \chi_{ijk}^2 \\
&= -\frac{1}{2} \sum_{i(jk)} \kappa_{ijk}^{\mathrm{CC}} \left[\mathbf{S}_i \cdot \left(\mathbf{S}_j \times \mathbf{S}_k\right)\right]^2 ,
\end{aligned}
\qquad (2)
$$

where the coefficient $\kappa_{ijk}^{\mathrm{CC}}$ describes the strength of the CCI within the triangle formed by $i$, $j$, and $k$.

Remarkably, this non-relativistic exchange interaction, which is quadratic in $\chi_{ijk}$, emerges without any external magnetic field as it is intrinsic to the ground state of complex spin textures. Symmetry allows for such type of interaction since time reversal inverts all spins and changes the sign of the chirality, but it keeps the energy of the spin system invariant. Adopting the quantum-mechanical viewpoint of ref. [32], the form of CCI can be also interpreted as a 6-spin–3-site interaction, which requires quantum mechanically speaking at least a spin-1 system.

**Spin-chiral interaction.** A valence electron experiencing the emerging orbital moment, $\mathbf{L}_i^{\mathrm{TO}}$, connected to its orbital motion around the $i$th ion in a non-coplanar texture is also exposed to the nuclear electric field acting as a magnetic field and coupling the orbital moment to its spin. Thus, this spin-chiral interaction (SCI) is a second element of the topological-chiral interaction, and it is a consequence of the relativistic SOI coupling the TOM to single spin magnetic moments, as illustrated in Fig. 1. The corresponding energy of the SCI can be shown (see Methods and Supplementary Note 1) to assume the form

$$
\begin{aligned}
E^{\mathrm{SC}} &= -\sum_i \kappa_i^{\mathrm{SC}} \mathbf{L}_i^{\mathrm{TO}} \cdot \mathbf{S}_i \\
&= -\sum_{i(jk)} \kappa_{ijk}^{\mathrm{SC}} \left(\boldsymbol{\tau}_{ijk} \cdot \mathbf{S}_i\right) \left[\mathbf{S}_i \cdot \left(\mathbf{S}_j \times \mathbf{S}_k\right)\right],
\end{aligned}
\qquad (3)
$$

where $\kappa_i^{\mathrm{SC}}$ is the spin-chiral coupling strength of magnetic atom $i$. Being linear in the scalar spin chirality, this relativistic 4th-order interaction is rotationally anisotropic.

Conceptually, the SCI energy as given by Eq. (2) is similar to the magneto-crystalline anisotropy energy (MAE), which prefers a maximal projection of the spin-orbit induced orbital moment onto the local spin quantization axis[33,34]. While the chiral-chiral coupling roots solely in the spin configuration, the spin-chiral coupling is very sensitive to the lattice structure, just like MAE and antisymmetric DMI exchange as discussed in Supplementary Note 1. In contrast to the MAE, however, the SCI is susceptible to the local chirality, which gives rise to its outstanding feature: the SCI energy favours states with a certain chirality as exemplified for real materials below. The SCI is also distinct from the recently-uncovered chiral biquadratic interaction, which generalizes the DMI to a four-spin interaction but on two sites[35]. Thus, the SCI is a novel type of magnetic interaction that can replace the DMI in stabilizing chiral ground states.

The form of Eqs. (1)–(3) is adapted to the case of cubic crystals for which the local constants $\kappa_{ijk}^{\mathrm{CC}}$, $\kappa_{ijk}^{\mathrm{TO}}$, and $\kappa_{ijk}^{\mathrm{SC}}$ are scalar quantities. In general, however, these constants are tensor quantities, and we provide generalized expressions for the proposed interactions as well as the extended spin Hamiltonian in Supplementary Notes 1, 2, and 3.

**Topological-chiral interactions in real materials.** Next, we substantiate the importance of the topological–chiral interactions taking the intensively scrutinized B20 magnet MnGe as one specific example. In contrast to FeGe that forms well-understood two-dimensional skyrmions with a radius of 70 nm, for MnGe

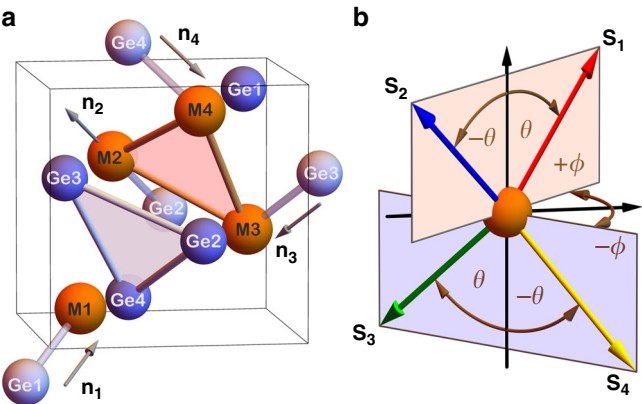

**Fig. 2 Crystal structure of the cubic B20 germanides with antiferromagnetic spin texture. a** The unit cell contains four magnetic (M = Fe, Mn) and four Ge (blue) atoms. The vector $\mathbf{n}_i$ indicates the local three-fold rotation axis on the $i$th atom type. **b** The particular antiferromagnetic spin configuration of the magnetic moments $\mathbf{S}_i$ used in this work is defined by $\theta$ and $\phi$.

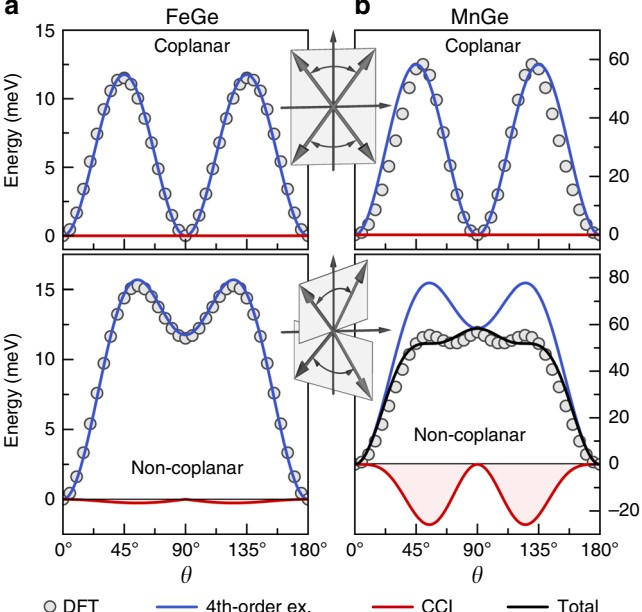

**Fig. 3 Beyond-Heisenberg interactions in B20 germanides.** Angular dependence of the total energy without spin-orbit coupling for coplanar ($\phi = 0°$) and non-coplanar ($\phi = 45°$) antiferromagnetic spin configurations in **a** FeGe and **b** MnGe. Circles denote the first-principles results whereas blue and red solid lines indicate the fitted 4th-order exchange and chiral-chiral (CCI) contributions, respectively. Note the difference of scales between **a** and **b**.

puzzling three-dimensional skyrmion lattices with lattice constants of a few nanometers were observed experimentally[13,18], but could not be reproduced by any theoretical calculation so far[36,37]. Employing electronic-structure theory (see Methods), we show below that while the chiral-chiral and spin-chiral interactions are suppressed in FeGe, where chiral physics is dominated by the Dzyaloshinskii–Moriya interaction (DMI)[15,16], the topological-chiral interactions are very prominent in MnGe.

The chiral structure of the cubic B20 compounds consists of four sub-lattices that are structurally identical, but their local three-fold rotation axes point along different cube diagonals $\mathbf{n}_i$, see Fig. 2a. We describe these systems by an effective spin-lattice Hamiltonian that includes Heisenberg and 4th-order exchange, DMI, magnetic anisotropy, and the proposed topological-chiral interactions, see Eq. (8) in the Methods section. To disentangle the different types of magnetic interactions, we consider a set of spin configurations which corresponds to one of the possible irreducible representations of the symmetry group of the paramagnetic phase (see Methods), and can be parameterized by

$$\mathbf{S}_i = \left( n_i^x \sin\theta\cos\phi, n_i^y \sin\theta\sin\phi, n_i^z \cos\theta \right), \quad (4)$$

where the unit vector $\mathbf{S}_i$ refers to the direction of the classical spin moment of atom type $i \in \{1, \dots, 4\}$. $\theta$ and $\phi$ are the polar and azimuthal angles, respectively, and $\mathbf{n}_i$ points along a cube diagonal, as illustrated in Fig. 2a, b.

This choice of magnetic structures greatly simplifies the analysis as the energy contribution of the classical Heisenberg interaction $(\mathbf{S}_i \cdot \mathbf{S}_j)$ among spins following Eq. (4) is independent of $\theta$ and $\phi$. Consequently, any non-trivial angular dependence of the total energy will indicate the presence of higher-order magnetic interactions. In fact, our first-principles calculations of the total energy in absence of spin-orbit coupling, see Fig. 3, show drastic energy variations with respect to $\theta$ and $\phi$ on the order of several tens of meV in both FeGe and MnGe, which discloses the absolute significance of beyond-Heisenberg terms in these materials. However, while 4th-order exchange interactions of the form $(\mathbf{S}_i \cdot \mathbf{S}_j)(\mathbf{S}_k \cdot \mathbf{S}_l)$[32] indeed describe the energy variation with $\theta$ well in FeGe, their contribution deviates strongly from the calculated curve in MnGe if the spins are not coplanar, $\phi = 0°$, but non-coplanar, e.g., for $\phi = 45°$ (see Supplementary Note 3 and Supplementary Table 1 for further details).

To uncover the distinct nature of these large qualitative discrepancies between 4th-order exchange and the computed

total-energy variation in non-coplanar MnGe, we consider the physically motivated CCI, Eq. (2), as given by the coupling of emergent orbital currents due to the complex spin topology. Taking into account the crystal symmetries of the B20 compounds and the spin configurations Eq. (4), we find that $E^{CC} = -2\kappa^{CC}[\sin\theta\sin 2\theta\sin 2\phi]^2$, with a single material-specific coupling constant $\kappa^{CC}$. Fitting this angular dependence to our ab initio results significantly improves the description of the total-energy variation, capturing now all essential features also for MnGe, Fig. 3b. Modelling additionally changes of the spin-moment length with $\theta$ enhances the agreement with our calculations even further (see Supplementary Note 4 and Supplementary Fig. 1). The coupling constant $\kappa^{CC}$ of the CCI amounts to 1.2 meV and 59.9 meV in FeGe and MnGe, respectively, which underlines the absolute relevance of the proposed interaction in the latter system. Since the spin moment is 0.78 $\mu_B$ in FeGe but amounts to 1.96 $\mu_B$ for MnGe, this finding of a significant energy contribution due to the chiral–chiral coupling is consistent with the quantum-mechanical mechanism for the CCI, which necessitates at least a spin-1 system.

By computing explicitly the orbital moment without SOI on each magnetic atom, see Fig. 4b, we further substantiate the distinct microscopic origin of the chiral–chiral coupling in MnGe. Based on the angular dependence of $\chi_{ijk}$ for the choice (4) of magnetic structures, Eq. (1) leads to $\mathbf{L}_i^{TO} = -\kappa^{TO}\sin\theta\sin 2\theta\sin 2\phi\,\mathbf{n}_i$ such that the TOM at the $i$th atom is collinear to the cube diagonal $\mathbf{n}_i$, and the magnitude is proportional to both the spin chirality and the single effective topological orbital susceptibility constant $\kappa^{TO}$. This analytic expression is in perfect agreement with the ab initio calculations of the local orbital moment in absence of SOI as shown in Fig. 4b, revealing that the topological orbital susceptibility is of opposite sign in the two compounds, namely, $\kappa_{FeGe}^{TO} = -0.02\,\mu_B$ and $\kappa_{MnGe}^{TO} = 0.13\,\mu_B$. Our results clearly demonstrate that the

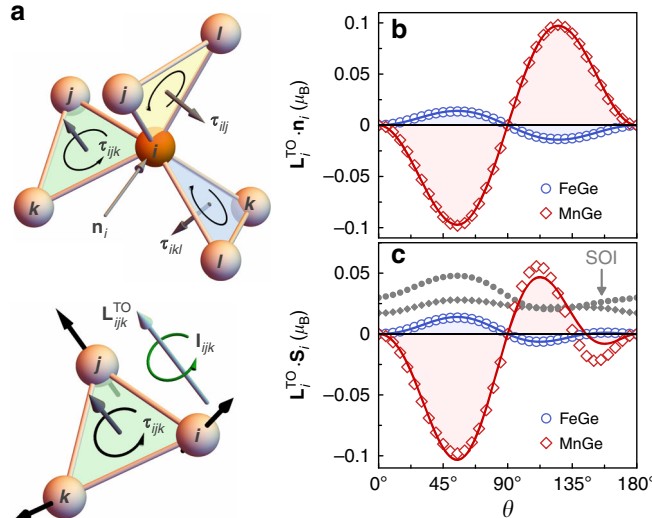

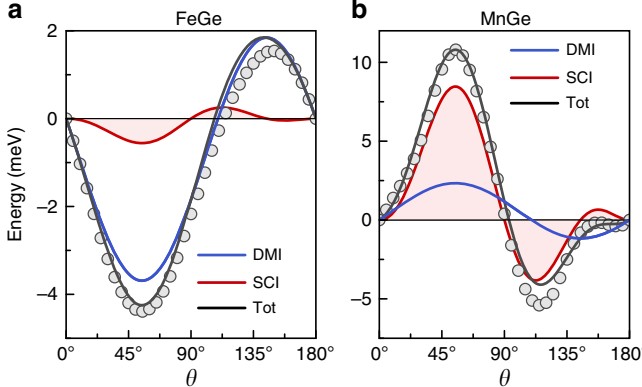

**Fig. 4 Topological orbital magnetism in B20 germanides. a** The central magnetic atom and its magnetic neighbours form three symmetry-related triangles with corresponding normal vectors $\tau_{ijk}$. Owing to the non-coplanar spin configuration of each such triangle, the finite persistent electron current $\mathbf{I}_{ijk}$, indicated by the green arrow, contributes to the orbital moment. **b**, **c** Projections of the resulting topological orbital moment $\mathbf{L}_i^{TO}$ of the $i$th atom type onto the local symmetry axis $\mathbf{n}_i$, and the local spin direction $\mathbf{S}_i$. Open symbols refer to first-principles results and solid lines are the model fit. In addition, we mark by small full symbols in **c** the ab initio SOI contribution to the orbital moment.

**Fig. 5 Spin-orbit mediated interactions in B20 germanides.** Comparing in **a** FeGe and **b** MnGe the spin-orbit-induced energy change for $\phi = 45°$ between the first-principles calculations (circles) and the model fit (lines), resolved in terms of antisymmetric exchange (DMI) and spin-chiral interaction(SCI).

TOM in MnGe, and specifically its variation with $\theta$, exceeds the effect in FeGe by roughly an order of magnitude, which underpins the importance of the predicted CCI in the former material.

While our predictions of isotropic 4th-order and chiral–chiral contributions to the exchange energy are independent of the spin–orbit coupling, we elucidate now the role of relativistic SOI for the angular dependence of the total energy. Although the magnetocrystalline anisotropy is uniaxial to lowest order in the non-collinear B20 magnets (see Supplementary Note 5), we explicitly verified that the corresponding energy contribution is negligibly small as compared to all other terms considered here.

Figure 4c displays the projection $(\mathbf{L}_i^{TO} \cdot \mathbf{S}_i)$ computed without consideration of the SOI, which enters the expression (3) for the spin–chiral coupling. The projection is generally much larger in MnGe than in FeGe, with a very pronounced dependence on the chiral spin texture. In sharp contrast, the additional corrections to the total orbital moment due to SOI exhibit only minor modulations with the angle $\theta$.

As a result of the small projection in case of FeGe, the spin-orbit correction of the total energy with respect to $\theta$ is primarily due to the antisymmetric exchange $(\mathbf{S}_i \times \mathbf{S}_j)$ as illustrated in Fig. 5a, whereas the substantial angular dependence in MnGe cannot be described solely by the DMI energy (see also Supplementary Note 3 and Supplementary Table 1). For MnGe, using Eq. (3), spin configuration (4) and a single constant $\kappa^{SC}$, we obtain the form $E^{SC} = 2\kappa^{SC}\kappa^{TO}\sin^2 2\theta \sin 2\phi [1 + \tan\theta (\cos\phi + \sin\phi)]$ for the energy contribution of the proposed SCI that substantially improves the modelling of the total-energy variations based on the first-principles results, see Fig. 5b. By directly fitting Eq. (3) to our ab initio data, we find that the constant $\kappa^{SC}\kappa^{TO}$ is 0.2 meV in FeGe, and $-3.2$ meV in MnGe. Thus, we conclude from our analysis that the novel SCI is by far the dominant spin-orbit effect in MnGe – contrary to FeGe, where physical properties are hardly affected by the TOM but driven rather by the prominent DMI.

**Emergence of three-dimensional textures**. For magnetization textures with characteristic length scales much larger than the underlying crystal lattice, a continuum representation, $\mathbf{S}_i \to \mathbf{m}(\mathbf{r})$, of the spin-lattice Hamiltonians (2) and (3) is able to reflect the essence of the magnetic interactions. Taking this micromagnetic approach, the relevant order parameter of the system is a magnetization density $\mathbf{m}(\mathbf{r}) = (m_x, m_y, m_z)$, $|\mathbf{m}(\mathbf{r})| = 1$, a unit-vector field, defined at any point $\mathbf{r} \in \mathbb{R}^3$. As shown in Supplementary Note 6 the scalar spin-chirality $\chi_{ijk}$ emerges in the continuum theory as a spin-chirality density, which is a dot product of a direction vector of a surface normal and the solenoidal gyro-vector field $\mathbf{F}(\mathbf{r})$, whose components $F_\alpha = \sum_{\beta\gamma} \varepsilon_{\alpha\beta\gamma} f_{\beta\gamma}$, are determined by the unit magnetization field, $\mathbf{m}$, and its spatial derivatives:

$$f_{\beta\gamma} = \mathbf{m} \cdot \left[\frac{\partial \mathbf{m}}{\partial r_\beta} \times \frac{\partial \mathbf{m}}{\partial r_\gamma}\right] \quad \text{with} \quad \alpha, \ \beta, \ \gamma, \ \in \{x, y, z\}, \qquad (5)$$

where $\varepsilon_{\alpha\beta\gamma}$ is the antisymmetric 3D Levi–Civita tensor. The gyro-vector field is linearly related to the TOM density. The micromagnetic expression for the energy of the CCI takes then the following form:

$$E_{mm}^{CC} = -\frac{1}{2}\tilde{\kappa}^{CC} \sum_{\alpha,\beta=1}^{3} \int d\mathbf{r} \, F_\alpha(\mathbf{r}) F_\beta(\mathbf{r}) \, , \qquad (6)$$

where the micromagnetic $\tilde{\kappa}^{CC}$ is proportional to the microscopic $\kappa^{CC}$ and it is assumed for simplicity to be a single constant. Together with the Heisenberg exchange, Eq. (6) describes the energy of the magnetization texture as a variant of the highly acclaimed Faddeev model[30], the exciting feature of which is that it contains hopfions, stable localized three-dimensional knotted topological solitons as solutions. The role of the SCI will be to orient the hopfion relative to the underlying lattice.

## Discussion

Opening several exciting vistas in the field of chiral magnetism, our findings raise a number of important fundamental questions. Above all, they call for a review of the relevance of the chiral-chiral and spin-chiral coupling, uncovered in this study, for the ground state of existing materials that exhibit diverse magnetic orders. This also concerns the systems in which emerging homochiral magnetic structures were previously thought to be the result of the Dzyaloshinskii–Moriya interaction (DMI). In

contrast, topological–chiral interactions offer fundamentally different opportunities for imprinting chiral magnetism, as they manifest in the scalar chirality of spin arrangements on triangular plaquettes, as opposed to the vector chirality between pairs of spins in the case of DMI. In the continuum limit, the spin-chirality relates to the curvature of the magnetization field and the chiral-chiral interaction reverts to the Faddeev model. Thus, magnets with topological-chiral magnetic interactions offer the first experimental realization of this model with hopfions as the emergent 3D magnetic particles. It could be speculated that the unique 3D magnetic order of MnGe explained in this article by the occurrence of the topological–chiral interaction is a precursor state. Another important aspect to be explored is the influence of the discovered interactions on the dynamical properties of ferromagnetic, chiral, and antiferromagnetic systems. As we show in Supplementary Note 7, the corresponding modifications to the phenomenological model for the free energy of antiferromagnets, brought about by topological–chiral interactions, enable a direct interpretation of magnetic phase transitions at high pressure and finite temperature. This provides a foundation for studying antiferromagnetic dynamics in materials that exhibit the proposed chiral-chiral and spin-chiral interactions.

## Methods

**First-principles calculations.** We used the FLEUR code (http://www.flapw.de) to calculate the total energy, spin and orbital magnetic moments of MnGe and FeGe in the B20 phase with and without spin-orbit interaction for a set of non-collinear magnetic structures within the generalized gradient approximation of Perdew–Burke–Ernzerhof[38]. The first Brillouin zone was sampled with a $12 \times 12 \times 12$ Monkhorst-Pack grid. To converge the energy calculations for the single-site magnetocrystalline anisotropy, the sampling of the full Brillouin zone was increased to $24 \times 24 \times 24$ **k**-points and the temperature in the Fermi distribution was reduced from 160 K to 80 K. The plane-wave cutoff for the basis functions was 4.2 a.u.$^{-1}$. The cubic lattice structure of the B20 compounds[39] consists of four sub-lattices, see Fig. 2a, with the ions located at $(u, u, u)$, $(0.5 - u, 1 - u, 0.5 + u)$, $(1 - u, 0.5 + u, 0.5 - u)$, and $(0.5 + u, 0.5 - u, 1 - u)$, where we optimized the structural parameter $u$ together with the lattice constant $a$ by minimizing the total energy. The obtained values are $u_{Fe} = 0.134$, $u_{Ge} = 0.842$ in FeGe with the lattice constant $a = 4.67$ Å, and $u_{Mn} = 0.136$, $u_{Ge} = 0.843$ in MnGe with $a = 4.76$ Å.

Following this scheme, we computed the energies and orbital moments for different non-collinear antiferromagnetic (AFM) configurations, as described by two angles $\theta$ and $\phi$:

$$\mathbf{S}_i = \begin{pmatrix} \sin\theta_i \cos\phi_i \\ \sin\theta_i \sin\phi_i \\ \cos\theta_i \end{pmatrix} = \begin{pmatrix} n_i^x \sin\theta \cos\phi \\ n_i^y \sin\theta \sin\phi \\ n_i^z \cos\theta \end{pmatrix}. \quad (7)$$

Here $\phi_i = \{\phi, \phi, -\phi, -\phi\}$, $\theta_i = \{\theta, -\theta, \pi + \theta, \pi - \theta\}$, and the vector $\mathbf{n}_i$ is the local symmetry direction at site $i$ that points along one of the cube diagonals $(1, 1, 1)$, $(-1, -1, 1)$, $(-1, 1, -1)$, and $(1, -1, -1)$, see Fig. 2a. The spin texture given by Eq. (4) forms one of the irreducible representations of possible AFM configurations, which can be classified according to the symmetry properties of the AFM order parameters (see Supplementary Note 7).

Depending on the spin configuration, the magnetic moments of Fe and Mn amount to about 0.78 $\mu_B$ and 1.96 $\mu_B$, respectively, see Supplementary Fig. 1. The tiny induced magnetic moment of Ge ($< 0.005\ \mu_B$) is given with respect to the local quantization axis, which was chosen to follow the magnetization direction of its nearest-neighbour magnetic ion. Based on the ab initio electronic structure, the orbital magnetic moment $\mathbf{L}_i$ is obtained by locally integrating the angular momentum operator $\mathbf{L} = \mathbf{r} \times \mathbf{p}$ within the muffin-tin sphere of radius 2.25 and 2.30 a.u. for Fe and Mn, respectively, centered around the $i$th nucleus. To extract the topological part $\mathbf{L}_i^{TO}$, we evaluate the orbital moment in the absence of spin–orbit coupling.

The changes to the energetics and the orbital moments due to the SOI have been included based on the self-consistent electronic structure without SOI, using the so-obtained vector spin density and including the SOI applying the force theorem[40]. For phenomenological modelling we applied the Landau theory of phase transitions and introduced antiferromagnetic order parameters based on the symmetry analysis. Further details of modelling are provided in Supplementary Note 7.

**Spin-lattice Hamiltonian.** For our study we considered the following effective spin-lattice Hamiltonian:

$$H = -\sum_{ij} J_{ij} \mathbf{S}_i \cdot \mathbf{S}_j - \sum_{ijkl} K_{ijkl} (\mathbf{S}_i \cdot \mathbf{S}_j)(\mathbf{S}_k \cdot \mathbf{S}_l)$$
$$- \sum_{ij} \mathbf{D}_{ij} \cdot (\mathbf{S}_i \times \mathbf{S}_j) - \sum_i \mathbf{S}_i \underline{A}_i \mathbf{S}_i^\dagger$$
$$- \sum_{i(jk)} \kappa_{ijk}^{CC} \left[ \mathbf{S}_i \cdot \left( \mathbf{S}_j \times \mathbf{S}_k \right) \right]^2 / 2$$
$$- \sum_{i(jk)} \kappa_{ijk}^{SC} \left( \boldsymbol{\tau}_{ijk} \cdot \mathbf{S}_i \right) \left[ \mathbf{S}_i \cdot \left( \mathbf{S}_j \times \mathbf{S}_k \right) \right], \quad (8)$$

where $J_{ij}$, $K_{ijkl}$, $\mathbf{D}_{ij}$, $\kappa_{ijk}^{CC}$, and $\kappa_{ijk}^{SC}$ mediate Heisenberg, symmetric 4th-order, Dzyaloshinskii–Moriya, chiral–chiral, and spin–chiral couplings, respectively. The tensor $\underline{A}_i$ encodes the single-site magnetic anisotropy at site $i$, see Supplementary Note 5, and the unit vector $\mathbf{S}_i$ refers to the direction of the classical spin moment of atom type $i \in \{1, \ldots, 4\}$. While the considered terms for CCI, SCI, and magnetic anisotropy in the above expression represent the dominant local contributions to the total energy, a more general form with non-local contributions is given in Supplementary Note 2.

**Multiple scattering theory.** We derived the expressions for the proposed topological-chiral magnetic interactions by a systematic expansion of the total energy of the many-electron system with respect to simultaneous infinitesimal rotations of the magnetic moments. By rotating the set of magnetic moments within the solid, their mutual interaction can be obtained from the change of the total energy. To extract the interaction parameters, we invoke a multiple scattering theory realized by the Korringa–Kohn–Rostoker (KKR) Green-function formalism. Explicit expressions for the rigorous derivation of the topological-chiral interactions are provided in Supplementary Note 1.

## Data availability

The data that support the findings of this study are available from the corresponding authors upon reasonable request.

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

## Acknowledgements

We acknowledge fruitful discussions with Christof Melcher. We acknowledge funding from Deutsche Forschungsgemeinschaft (DFG) through SPP 2137 "Skyrmionics", the Collaborative Research Centers SFB 1238 and SFB/TRR 173, project MO 1731/5-1 and project SHARP 397322108. M.H., Y.M, and S.B. acknowledge the DARPA TEE program through grant MIPR# HR0011831554 from DOI, and O.G. acknowledges the EU FET Open RIA Grant no. 766566, Humboldt Foundation, and the ERC Synergy Grant SC2 (No. 610115). S.L. and J.B. express gratitude to the European Research Council (ERC) under the European Union's Horizon 2020 research and innovation program for funding (ERC-consolidator grant 681405 – DYNASORE). Simulations were performed with computing resources granted by JARA-HPC from RWTH Aachen University and Forschungszentrum Jülich under projects jara0161, jiff40 and jias1a.

## Author contributions

S.B. and Y.M. motivated the project. S.G., G.B., Y.M. and S.B. devised the details of the project. S.G. uncovered the role of chirality for the magnetic interactions by performing the DFT calculations with assistance from M.H., G.B. and J.-P.H. The expressions for the proposed interactions (2) and (3) and their corresponding physical picture were developed by S.G. and Y.M. The rigorous derivation of the proposed interactions in terms of Green functions was worked out by J.B. and S.L. The phenomenological model was developed by O.G. and S.G. The manuscript and the supplement were written by J.-P.H., S.G., M.H., S.L., Y.M. and S.B. All the authors contributed to the analysis and interpretation of the results.

## Competing interests

The authors declare no competing interests.
