## [Peer Review File · Nature Communications]

Reviewers' comments:

Reviewer #1 (Remarks to the Author):

A magnetic interaction stabilizing magnetic structures with finite chirality is studied theoretically. The interaction, as far as I know, has not been considered so far, and thus the study has novelty. The finding that the interaction can realize new chiral structures different from those due to DMI may be interesting for experts. I cannot, however, recommend publication in Nature Communications in the present form because of the following issues.

1. The proposed interaction, Eq. (2), is the one including 6 spins. Such higher order interaction would not be relevant compared to other lower order contributions, which are in fact totally neglected in the study. As the authors mention in the text, the interaction is a type of magnetic anisotropy energy, and there are numbers of different forms of the interaction allowed if up to 6 spins are involved. The authors need to justify their analysis picking up only one particular (beautiful) interaction and neglecting all the others. Eq. (4), which the authors argue to be small, would be one of the many possibilities.
2. The text starting from the next paragraph of Eq. (2) is not clearly-written, and what the authors want to discuss is difficult to see. For instance, TOM, which most readers would not know, is not defined and because of this, what Eq. (3) indicates or how important the relation (3) is are not perhaps understandable for most readers.
3. Most importantly, the study would not be of broad interest. Shift of the tilting angle from like 150 to 110 degree (Fig. 4) for MnGe by introducing the new interaction does not sound significant.

Reviewer #2 (Remarks to the Author):

In this manuscript, the authors calculated, using first-principles calculations, spin-spin interactions in representative B20 compounds FeGe and MnGe. Very interesting interactions, the chiral-chiral and spin-chiral interactions, are identified in MnGe. This report is exciting. Particularly the chiral-chiral interaction can possibly realize the Fadeev model and lead to exotic three-dimensional hopfion structure. I would like to recommend this manuscript to Nature Communications as long as the following minor concern is addressed.

The presence of CCI in MnGe is confirmed by extracting the 4th order spin exchange energy from the total energy. However, the authors must consider onsite crystalline magnetoanisotropy. In such cubic material, maybe 2nd order anisotropy is vanishing, but 4th order and 6th order anisotropy exists. Particularly, it was understood that in MnGe, the 4th order magnetoanisotropy is crucial, so that the superposition of three orthogonal spin helices leads to the monopole lattice therein. Therefore the authors must fit parameters by taking into account all these equal or lower order interactions compared to the chiral-chiral interaction. I am not sure whether chiral-chiral interaction is still significant in this way.

A minor correction. In the sentence 'In the absence of SOI, hopping in a non-coplanar texture with finite $\chi_{ijk} = S_i \cdot (S_j \times S_k)$ is equivalent to the electron dynamics in a coplanar spin background but in the presence of a fictitious magnetic field', the second 'coplanar' should be 'collinear' if my understanding is right.

In conclusion, that is an excellent manuscript that I would like to recommend potentially.

Reviewer #3 (Remarks to the Author):

The authors consider the spin-spin interactions in the B20 crystal structure materials MnGe and FeGe. They find that there are also more complex interactions between the spins. These new terms are of the form of couplings between topological orbital moments and interactions between topological orbital moments and with the lattice spins. These topological interactions are particularly strong in MnGe.

These features could be the origin of unresolved spin structures in this material. In the continuum limit, the couplings between the orbital moments resemble a version of the Faddeev model that contains hopfions, three-dimensional solitons. The realization of such structures would attract considerable attention in the magnetism community.

The manuscript describes a thorough, intriguing, and most likely sound calculation of the spin-spin interactions in B20 crystal magnetic systems. However, the presentation and nature of the research are quite technical, which, I believe, would prevent the attention of a broad audience.

I, therefore, recommend resubmission to a more specialized journal in condensed matter physics.

1 Reviewers' comments

Reviewer # 1

A magnetic interaction stabilizing magnetic structures with finite chirality is studied theoretically. The interaction, as far as I know, has not been considered so far, and thus the study has novelty. The finding that the interaction can realize new chiral structures different from those due to DMI may be interesting for experts. I cannot, however, recommend publication in Nature Communications in the present form because of the following issues.

1. The proposed interaction, Eq. (2), is the one including 6 spins. Such higher order interaction would not be relevant compared to other lower order contributions, which are in fact totally neglected in the study. As the authors mention in the text, the interaction is a type of magnetic anisotropy energy, and there are numbers of different forms of the interaction allowed if up to 6 spins are involved. The authors need to justify their analysis picking up only one particular (beautiful) interaction and neglecting all the others. Eq. (4), which the authors argue to be small, would be one of the many possibilities.
2. The text starting from the next paragraph of Eq. (2) is not clearly-written, and what the authors want to discuss is difficult to see. For instance, TOM, which most readers would not know, is not defined and because of this, what Eq. (3) indicates or how important the relation (3) is are not perhaps understandable for most readers.
3. Most importantly, the study would not be of broad interest. Shift of the tilting angle from like 150 to 110 degree (Fig. 4) for MnGe by introducing the new interaction does not sound significant.

Reviewer # 2

In this manuscript, the authors calculated, using first-principles calculations, spin-spin interactions in representative B20 compounds FeGe and MnGe. Very interesting interactions, the chiral-chiral and spin-chiral interactions, are identified in MnGe. This report is exciting. Particularly the chiral-chiral interaction can possibly realize the Fadeev model and lead to exotic three-dimensional hopfion structure. I would like to recommend this manuscript to Nature Communications as long as the following minor concern is addressed.

1. The presence of CCI in MnGe is confirmed by extracting the 4th order spin exchange energy from the total energy. However, the authors must consider onsite crystalline magnetoanisotropy. In such cubic material, maybe 2nd order anisotropy is vanishing, but 4th order and 6th order anisotropy exists. Particularly, it was understood that in MnGe, the 4th order magnetoanisotropy is crucial, so that the superposition of three orthogonal spin helices leads to the monopole lattice therein. Therefore the authors must fit parameters by taking into account all these equal or lower order interactions compared to the chiral-chiral interaction. I am not sure whether chiral-chiral interaction is still significant in this way.
2. A minor correction. In the sentence "In the absence of SOI, hopping in a non-coplanar texture with finite $\chi_{ijk} = \mathbf{S}_i \cdot (\mathbf{S}_j \times \mathbf{S}_k)$ is equivalent to the electron dynamics in a coplanar spin background but in the presence of a fictitious magnetic field", the second "coplanar" should be "collinear" if my understanding is right.

In conclusion, that is an excellent manuscript that I would like to recommend potentially.

Reviewer # 3

The authors consider the spin-spin interactions in the B20 crystal structure materials MnGe and FeGe. They find that there are also more complex interactions between the spins. These new terms are of the form of couplings between

topological orbital moments and interactions between topological orbital moments and with the lattice spins. These topological interactions are particularly strong in MnGe.

These features could be the origin of unresolved spin structures in this material. In the continuum limit, the couplings between the orbital moments resemble a version of the Faddeev model that contains hopfions, three-dimensional solitons. The realization of such structures would attract considerable attention in the magnetism community.

The manuscript describes a thorough, intriguing, and most likely sound calculation of the spin-spin interactions in B20 crystal magnetic systems. However, the presentation and nature of the research are quite technical, which, I believe, would prevent the attention of a broad audience. I, therefore, recommend resubmission to a more specialized journal in condensed matter physics.

2 Response to Reviewers

2.1 Overview of changes based on reviewers' comments:

1. We restructured our manuscript to highlight the physical origin of the discovered interactions more clearly, with particular emphasis on their relevance for realizing three-dimensional magnetic solitons. In this way, we further compactified the technical details and increased overall readability.
2. We revised Fig. 1 to underline the intuitive picture behind the proposed interactions.
3. We modified the paragraphs on the topological orbital moment to promote its origin and its relation to the novel exchange couplings.
4. We added a Supplementary Note on the single-ion anisotropy in B20 magnets. In addition, one sentence about the minor role of this anisotropy as compared to the predicted interactions is added to the main text.

2.2 Reviewer # 1

We would like to thank the Reviewer for the critical reading as well as for providing valuable remarks, which helped us to improve the manuscript. In the following, we present our point-by-point response to all raised issues.

A magnetic interaction stabilizing magnetic structures with finite chirality is studied theoretically. The interaction, as far as I know, has not been considered so far, and thus the study has novelty. The finding that the interaction can realize new chiral structures different from those due to DMI may be interesting for experts. I cannot, however, recommend publication in Nature Communications in the present form because of the following issues.

Reply: We thank the Reviewer for pointing out that our “study has novelty” and that it “may be interesting”. We share the opinion that this work is interesting for experts in the field of magnetism, materials science, spintronics, neuromorphic computing and those experts interested in the fundamental understanding of interactions in solids in general. We all know that new interactions with promising properties motivate materials scientists to come up with materials where these properties are honed and can be used in novel functional devices that can have great impact on our society. The last papers putting forward a novel magnetic interaction, the ones by Moriya, are cited today more than 4200 times by a large spectrum of experts. In this context, our work opens a new path towards realizing new classes of complex magnetic materials and textures, as we outline in our response to comment 3.

Comment 1: The proposed interaction, Eq. (2), is the one including 6 spins. Such higher order interaction would not be relevant compared to other lower order contributions, which are in fact totally neglected in the study. As the authors mention in the text, the interaction is a type of magnetic anisotropy energy, and there are numbers of different forms of the interaction allowed if up to 6 spins are involved. The authors need to justify their analysis picking up only one particular (beautiful) interaction and neglecting all the others. Eq. (4), which the authors argue to be small, would be one of the many possibilities.

Reply: We thank the Reviewer for this remark. First of all, we would like to stress that lower-order interactions are by no means “totally neglected” in our study. In addition to the new exchange couplings that we propose, in fact, we include explicitly the second-order Heisenberg and Dzyaloshinskii-Moriya terms as well as well-known fourth-order interactions in our work. This is also reflected in our manuscript, for example, on pages 4 and 5 of the main text, and in Supplementary Note 6, where we provide the angular dependencies of these conventional interactions together

with the coupling parameters. In addition, we provide in our revised manuscript a Supplementary Note that discusses the minor role of the magnetocrystalline anisotropy (see also our response to Reviewer 2).

Of course, we agree with the Reviewer that very different forms of the interaction terms are allowed if many spins are involved. However, we do not “pick up” the proposed interactions at random but we develop a clear interpretation for the underlying form using an intuitive physical picture (see also our response to comment 2). This allows us to identify those terms that are motivated purely by physics instead of mathematical combinatorics. To convey this aspect more clearly, we substantially restructured our revised manuscript, emphasizing now the physical origin of the discovered exchange interactions. Our results, shown in Figs. 2 and 4 of the main text, demonstrate that this microscopic intuition succeeds in describing excellently the physics in real magnetic materials.

Finally, we point out that the interaction described by Eq. (4) in the original manuscript (which is Eq. (3) in the revised text) is by no means small in the presence of spin-orbit coupling. In fact, the coupling given by this equation, which we term spin-chiral interaction, can dominate over the well-known Dzyaloshinskii-Moriya interaction and thereby determines the magnetic ground state. We explicitly demonstrate this utter relevance of the spin-chiral interaction in Fig. 4b of the main text for the example of MnGe.

Comment 2: The text starting from the next paragraph of Eq. (2) is not clearly-written, and what the authors want to discuss is difficult to see. For instance, TOM, which most readers would not know, is not defined and because of this, what Eq. (3) indicates or how important the relation (3) is are not perhaps understandable for most readers.

Reply: We apologize that we could not convey the intuitive physical picture that underlies the proposed interactions with sufficient clarity such that the Reviewer can appreciate this line of thought. Accordingly, we revised the corresponding paragraphs in the main text and moved them to the beginning of the Results section to clarify the origin of the topological orbital moment (TOM) and its relation to the new exchange couplings. However, we would like to point out that Eq. (3) in the original manuscript (which is Eq. (1) in the revised manuscript) provides already the mathematical definition of the TOM, which stems from the electron’s motion in a non-collinear magnetic structure. The latter acts as an emergent magnetic field that modifies the static and dynamical properties of the electronic states. The coupling of the TOM, either to the emergent magnetic field or to the spin structure, manifests in the proposed interactions. We illustrate this mechanism also in the revised Fig. 1.

Comment 3: Most importantly, the study would not be of broad interest. Shift of the tilting angle from like 150 to 110 degree (Fig. 4) for MnGe by introducing the new interaction does not sound significant.

Reply: We cannot agree with the Reviewer’s opinion that our work “would not be of broad interest”. On the contrary, we are convinced that our work is of broad interest to a general readership in magnetism as we uncover qualitatively new, intuitive interactions that change the ground state of magnetic materials. Specifically, the proposed spin-chiral interaction can completely replace the Dzyaloshinskii-Moriya interaction in selecting a chiral ground state. As a consequence, we anticipate that these findings will trigger the review of established observations of complex magnetic ground states in the light of the proposed chiral interactions. Even more importantly, since these interactions realize an elusive mathematical model that hosts unique soliton solutions, our work advances the research on innately three-dimensional magnetic textures offering great potential for future information processing. Therefore, the conceptual scope of the proposed interactions goes well beyond the studied examples of B20 materials, which we used as proof-of-principle, and our work will thereby impact the field of chiral magnetism in general.

2.3 Reviewer # 2

We would like to thank the Reviewer for the critical reading as well as for providing very insightful remarks, which really helped us to improve the manuscript. In the following, we present our point-by-point response to all raised issues.

In this manuscript, the authors calculated, using first-principles calculations, spin-spin interactions in representative B20 compounds FeGe and MnGe. Very interesting interactions, the chiral-chiral and spin-chiral interactions, are identified in MnGe. This report is exciting. Particularly the chiral-chiral interaction can possibly realize the Fadeev model and lead to exotic three-dimensional hopfion structure. I would like to recommend this manuscript to Nature Communications as long as the following minor concern is addressed.

Reply: We thank the Reviewer for considering our work as “exciting” and the proposed interactions as “very interesting”. We hope that we have addressed the “minor concern” adequately such that the Reviewer can recommend publication in Nature Communications.

Comment 1: The presence of CCI in MnGe is confirmed by extracting the 4th order spin exchange energy from the total energy. However, the authors must consider onsite crystalline magnetoanisotropy. In such cubic material, maybe 2nd order anisotropy is vanishing, but 4th order and 6th order anisotropy exists. Particularly, it was understood that in MnGe, the 4th order magnetoanisotropy is crucial, so that the superposition of three orthogonal spin helices leads to the monopole lattice therein. Therefore the authors must fit parameters by taking into account all these equal or lower order interactions compared to the chiral-chiral interaction. I am not sure whether chiral-chiral interaction is still significant in this way.

Reply: We thank the Reviewer for this very insightful comment on the importance of the local magnetocrystalline anisotropy, which we are fully aware of. In fact, we have studied the role of the local anisotropy in these materials but decided originally not to include these results and conclusions into the submitted manuscript in order not to overburden the manuscript and de-focus the reader. However, considering the Reviewer’s valid remark, we are motivated to comment on this property in an additional Supplementary Note to the revised manuscript. In addition, we also added one sentence on this aspect in the main text. As a consequence of our analysis, we can demonstrate that the proposed exchange interactions remain of significant importance even if the local magnetic anisotropy is taken into account.

Comment 2: A minor correction. In the sentence ‘In the absence of SOI, hopping in a non-coplanar texture with finite $\chi_{ijk} = \mathbf{S}_i \cdot (\mathbf{S}_j \times \mathbf{S}_k)$ is equivalent to the electron dynamics in a coplanar spin background but in the presence of a fictitious magnetic field’, the second ‘coplanar’ should be ‘collinear’ if my understanding is right.

Reply: The Reviewer is right in that normally in the literature the fictitious magnetic field picture is applied to collinear ferromagnets. However, this does not have to be so. Within the picture of slowly varying spin textures, the emergent field technically arises due to the fictitious vector potential $\mathcal{A}(\mathbf{r}) = -iU(\mathbf{r})\nabla U(\mathbf{r})$ where position-dependent unitary matrices $U(\mathbf{r})$ rotate the non-collinear spins into the collinear configuration, see e.g. discussion around Eqs. (27)–(32) of Fujita et al. J. Appl. Phys. 110, 121301 (2011). By an easy analysis of this derivation it becomes immediately clear that the fact that the spins are rotated into a collinear state or another non-collinear configuration, plays no role at all, since the vector potential is determined only by the unitary spin rotations $U(\mathbf{r})$. In our case, discussed in this work, we start with a spin configuration which deviates slightly from a coplanar non-collinear configuration of 4 spins, and apply unitary rotations $U(\mathbf{r})$ to turn this state into the coplanar state. The slightly non-coplanar state (characterized with a value of scalar spin chirality χ_0) can be thus treated as a coplanar case with an emergent field applied to it. However, it is easy to realize that the very same rotation matrices, when applied to the *collinear AFM state* of 4 spins, will result in a slightly non-collinear state which has the very same value of chirality χ_0 . In other words, both slightly non-coplanar states can be treated as two initial states to which the very same emergent field is applied, with the first initial state being collinear AFM, and the second being coplanar non-collinear state. This is the reason why we can generally make the statement in the aforementioned sentence generally about coplanar configurations.

In conclusion, that is an excellent manuscript that I would like to recommend potentially.

Reply: We are grateful for the Reviewer’s assessment that our manuscript is “excellent”, and we hope that we have addressed adequately the raised points such that the Reviewer can recommend our revised manuscript for publication.

2.4 Reviewer # 3

We would like to thank the Reviewer for the critical reading of the manuscript as well as for providing valuable remarks, which helped us to improve the manuscript. In the following, we present our point-by-point response to all raised issues.

The authors consider the spin-spin interactions in the B20 crystal structure materials MnGe and FeGe. They find that there are also more complex interactions between the spins. These new terms are of the form of couplings between topological orbital moments and interactions between topological orbital moments and with the lattice spins. These topological interactions are particularly strong in MnGe. These features could be the origin of unresolved spin structures in this material. In the continuum limit, the couplings between the orbital moments resemble a version of the Faddeev model that contains hopfions, three-dimensional solitons. The realization of such structures would attract considerable attention in the magnetism community.

Reply: We thank the Reviewer for the excellent summary of the main content and messages of our work and its major consequences in the context of chiral magnetism and three-dimensional solitons. Specifically, we are grateful to the Reviewer for the assessment that realizing the latter particle-like magnetic configurations, for example, via the proposed exchange interactions, “would attract considerable attention in the magnetism community”.

The manuscript describes a thorough, intriguing, and most likely sound calculation of the spin-spin interactions in B20 crystal magnetic systems. However, the presentation and nature of the research are quite technical, which, I believe, would prevent the attention of a broad audience. I, therefore, recommend resubmission to a more specialized journal in condensed matter physics.

Reply: We are very sorry, but cannot really agree with the Reviewer that our work describes only the details of an “intriguing” calculation of spin interactions in B20 materials or that “the presentation and nature of the research are quite technical”. We have a different view on this issue.

In general, there are two kinds of scientific revolutions, those driven by new tools and those driven by new concepts. Our work belongs to the second category. The main focus of our manuscript is to introduce conceptually new chiral exchange interactions, that have been left unnoticed so far, that are of general importance and for which we develop an intuitive physical understanding. We use the material-specific electronic-structure calculations based on density functional theory (DFT) as a proof-of-principle to substantiate our concepts for real materials. We use these DFT calculations in replacement of an experiment. We have chosen MnGe as test system and we not only show that the interaction is there, we also show this interaction unexpectedly large and scientifically decisive. All computational details of these calculations necessary to repeat the calculations, we have put aside from the main paper into the Supplementary Note. To convince the scientific community of our finding, we find it necessary to explicitly and briefly explain the community how we disentangled the microscopic signature of the proposed interactions from other known exchange couplings, and thus we present in the main text the minimal amount of the computational details. But we share the enthusiasm of the Reviewer, that the calculations carried out for the non-trivial B20 compounds as such are intriguing, considering the energy path that we have chosen. Nevertheless, taking into account the Reviewer’s concern, we have restructured our manuscript, presenting now first the physical arguments that give rise to the proposed interactions before proving their importance in real materials. In this way, we could compactify the brief description of the calculation scheme to further increase readability.

As the Reviewer seems to have understood all of our main points according to the above summary, we are convinced that our work is, indeed, very understandable for the general community.

REVIEWERS' COMMENTS:

Reviewer #1 (Remarks to the Author):

The authors answered to the comments I gave. The beginning of the result section was improved, which makes readers easier to understand the message of the paper. However, I still think that the explanations on the system the authors assume could be much improved. For instance, the information on the interaction assumed is scattered in the latter half of the Result section and not easy to see as a whole. I wonder such an important equation should be in the main text instead of Supplementary Note 6. I believe the manuscript would be much better if written as a long self-contained article in other journals. The broad importance is always an issue of debate but I agree that the manuscript is above the criterion. Nevertheless, those points are not serious ones which prevent publication in Nature Communications. I recommend publication.

Reviewer #2 (Remarks to the Author):

I have carefully read the reply to all reviewers and the revised manuscript. I still believe that this work represents a frontier in the study of topological magnetism and could possibly open the door to topological and other three-dimensional topological spin textures. In the current manuscript, additional data on magnetoanisotropy is presented upon my request. I now agree with the authors that its effect is negligible due to its small value, which is even possibly smaller than the accuracy of DFT calculation. To conclude, I recommend this manuscript to Nature Communications.

1 Reviewers' comments

Reviewer # 1

The authors answered to the comments I gave. The beginning of the result section was improved, which makes readers easier to understand the message of the paper. However, I still think that the explanations on the system the authors assume could be much improved. For instance, the information on the interaction assumed is scattered in the latter half of Result section and not easy to see as a whole. I wonder such important equation should be in the main text instead of Supplementary Note 6. I believe the manuscript would be much better if written as a long self-contained article in other journals. The broad importance is always an issue of debate but I agree that the manuscript is above the criterion. Nevertheless, those points are not serious ones which prevent publication in Nature Communications. I recommend publication.

Reviewer # 2

I have carefully read reply to all reviewers and revised manuscript. I still believe that this work represents frontier in the study of topological magnetism and could possibly open the door to hopfion and other three dimensional topological spin textures. In the current manuscript, additional data on magnetoanisotropy is presented upon my request. I now agree with the authors that its effect is negligible due to its small value, which is even possibly smaller than the accuracy of DFT calculation. To conclude, I recommend this manuscript to Nature Communications.

2 Response to Reviewers

2.1 Reviewer # 1

The authors answered to the comments I gave. The beginning of the result section was improved, which makes readers easier to understand the message of the paper. However, I still think that the explanations on the system the authors assume could be much improved. For instance, the information on the interaction assumed is scattered in the latter half of Result section and not easy to see as a whole. I wonder such important equation should be in the main text instead of Supplementary Note 6. I believe the manuscript would be much better if written as a long self-contained article in other journals. The broad importance is always an issue of debate but I agree that the manuscript is above the criterion. Nevertheless, those points are not serious ones which prevent publication in Nature Communications. I recommend publication.

Reply: We are happy that we succeeded in further improving the clarity of our manuscript based on the helpful comments that we received. Reviewer 1 states clearly that our paper is above the criterion for publication in Nature Communications.

The reviewer debates whether it wouldn't be much better if the manuscript "is written as a long self-contained article in an other journal". We understand the wish of the Reviewer, but we argue that the invention of Appendices and Supplementary Materials are stylistic concepts to enhance the readability of manuscripts by separating important lines of thoughts from technicalities that are of course essential for the understanding and reproducibility of the content of the paper. This is particularly useful for the broader community and readers that are not directly at the core and at the nitty gritty details of the field, but interested in the greater context, while theoreticians and other experts deep in the field are typically interested in the details of the derivation. A longer paper contains the same content only differently distributed. Sometimes these longer papers with too many derivations in the main text shy experimentalists away. Therefore, we leave it in principle like this, but made efforts to optimise in the direction of the reviewer's suggestion subject to the space limit of the main text of 5000 words.

That means, even though the Reviewer recommends this work for publication in Nature communication without any changes, we found his statement about a whole picture of the magnetic interactions revealed in this studied too scattered across the paper serious and a relevant point to change the main text. Therefore, we included the full expression of the assumed Hamiltonian, which appeared before only in Supplementary Note 6, into the Methods section of the main text. As a consequence, we are convinced that the article is accessible more easily since the forms of all magnetic interactions are directly available to the reader. In addition, we modified accordingly the description in the Results section.

Finally, we would like to thank the Reviewer for providing valuable comments and for recommending our manuscript for publication.

2.2 Reviewer # 2

I have carefully read reply to all reviewers and revised manuscript. I still believe that this work represents frontier in the study of topological magnetism and could possibly open the door to hopfion and other three dimensional topological spin textures. In the current manuscript, additional data on magnetoanisotropy is presented upon my request. I now agree with the authors that its effect is negligible due to its small value, which is even possibly smaller than the accuracy of DFT calculation. To conclude, I recommend this manuscript to Nature Communications.

Reply: We are grateful to the Reviewer for carefully reading our manuscript and for the Reviewer's assessment that our work represents a "frontier in the study of topological magnetism" and that it could "open the door to hopfion

and other three dimensional topological spin textures”. We thank the Reviewer for the insightful comments during the review process and for recommending our manuscript for publication.